# Kozak Similarity Score Algorithm Identifies Alternative Translation Initiation Codons Implicated in Cancers

**DOI:** 10.3390/ijms231810564

**Published:** 2022-09-12

**Authors:** Alec C. Gleason, Ghanashyam Ghadge, Yoshifumi Sonobe, Raymond P. Roos

**Affiliations:** Department of Neurology, University of Chicago Medical Center, Chicago, IL 60637, USA

**Keywords:** translation initiation, canonical and noncanonical translation initiation codons, protein translation, oncogene, oncogenesis, tumorigenesis, cancer

## Abstract

Ribosome profiling and mass spectroscopy have identified canonical and noncanonical translation initiation codons (TICs) that are upstream of the main translation initiation site and used to translate oncogenic proteins. There have previously been conflicting reports about the patterns of nucleotides that surround noncanonical TICs. Here, we use a Kozak Similarity Score algorithm to find that nearly all of these TICs have flanking nucleotides closely matching the Kozak sequence. Remarkably, the nucleotides flanking alternative noncanonical TICs are frequently closer to the Kozak sequence than the nucleotides flanking TICs used to translate the gene’s main protein. Of note, the 5′ untranslated region (5‘UTR) of cancer-associated genes with an upstream TIC tend to be significantly longer than the same region in genes not associated with cancer. The presence of a longer-than-typical 5′UTR increases the likelihood of ribosome binding to upstream noncanonical TICs, and may be a distinguishing feature of a number of genes overexpressed in cancer. Noncanonical TICs that are located in the 5′UTR, although thought by some to be disadvantageous and suppressed by evolution, may translate oncogenic proteins because of their flanking nucleotides.

## 1. Introduction

Ribosome profiling and mass spectroscopy have demonstrated translation initiation within annotated regions in the human genome as well as outside of this region [1,2,3,4,5,6]. Codons that initiate translation in these positions may differ from the typical ATG sequence [7,8,9,10]. A recent study investigating a model of SOX2, which is inducibly expressed in oncogenic RAS-associated cancers, showed that translation initiation is upregulated at these unconventional upstream sites [11]. This noncanonical translation may express oncogenic proteins, thereby leading to tumor formation [11].

We previously introduced the Kozak Similarity Score (KSS) as a metric to compare nucleotides flanking a putative initiation codon with the Kozak sequence that surrounds an optimal ATG TIC [12]. The algorithm of KSS includes ten nucleotides preceding and following the codon:(1)$$KSS(codon)=\frac{1}{KSS\_bits_{\max}}\sum_{p=1}^{20}bits(nucleotide_{p})$$

In this equation, *p* denotes the position of a nucleotide bordering the codon. Values *p* = *1*, *2*, *3*, …, *10* designate the positions of the ten nucleotides (from left to right) on the left side of the codon, whereas values *p* = *11*, *12*, *13*, …, *20* designate the positions of ten nucleotides (from left to right) on the right side of the codon. Furthermore, *bits* is the assigned height of a particular nucleotide with reference to the Kozak sequence logo, related to the observed probability for a particular nucleotide to be at a certain position, as well as the impact of the position on the efficiency of translation initiation. *KSS*_*bits*_max_ is the maximum possible value of $\sum_{p=1}^{20}bits(nucleotide_{p})$.

KSS values are positively correlated with the likelihood of a canonical or noncanonical codon to initiate translation [12]. This is noteworthy, as the patterns of nucleotides that flank noncanonical TICs have been debated, and sequences that do not resemble the Kozak sequence have been proposed [13]. In the present study, we assess the ability of the KSS scoring system to identify noncanonical and canonical TICs implicated in the translation of oncogenic proteins. KSS effectively identifies these initiating codons, especially noncanonical TICs. Using experimental data from ribosome profiling and mass spectroscopy, we quantify the similarity of sequences surrounding upstream noncanonical TICs to the Kozak consensus motif.

## 2. Results

### 2.1. Canonical TICs That Translate the Main Protein as Well as Upstream Noncanonical TICs in Genes Associated with Cancer

We reviewed data from ribosome profiling and mass spectroscopy upstream of TICs used to translate the main protein from annotated genes [14]. Of note, ribosome profiling can predict TIC location on a large scale, while mass spectroscopy can confirm whether such TICs indeed induce measurable protein [15]. Although there are limitations to both methods of TIC identification [14,16,17], cross-verification by the two techniques is valuable. While there exist large public repositories of ribosome profiling data, from which a standardized canonical TIC library has been compiled [15], the same is unavailable for noncanonical TICs—most likely due to “noise” in noncanonical ribo-seq datasets.

A total of 101 unique TICs, of which the majority are noncanonical, were initially retrieved along with their mRNA sequences (see Methods). To find trends more specific for noncanonical TICs, we excluded seven instances in which the upstream TIC was an ATG. As a result, 94 unique noncanonical TICs (Appendix A) in 87 genes (Appendix A) were identified. Although these genes were not investigated for links to cancer in the original publication [14], we found that all but one gene are associated with oncogenesis, and all but seven of the genes (that are not well studied) are overexpressed in cancer. All of the TICs of the 87 genes were near-cognate, i.e., differing from ATG by only one nucleotide. Eighty of the 94 TICs initiate an N-terminally-extended variant of the main protein, with 12 initiating novel upstream proteins, and 2 initiating novel downstream proteins. All of the TICs were present in the 5′UTR except for two genes, *EPB41L3* and *SEPTIN9* (Figure 1). These two genes, which are associated with cancer [18,19,20,21], have noncanonical TICs *downstream* and not upstream from the gene’s canonical TIC.

We calculated the KSS for: a) ATG TICs used by the 87 genes for conventional translation of the main protein, b) upstream noncanonical TICs used for translation of genes associated with cancer (Figure 2). As a baseline for comparison, we calculated the KSS of all codons of the 87 gene transcripts that amounted to ~364,000, with one transcript per gene. In addition, 100,000 randomly chosen codons from 25,000 randomly chosen human mRNA sequences were obtained from the NCBI Nucleotide database (that contains sequences from other sources, including the GenBank [22], RefSeq [23], TBA, and PBD [24] databases). The baseline distribution had a median KSS of 0.58. Of note, the noncanonical TIC distribution was left-skewed, with a median KSS of 0.794. Although a small proportion of randomly chosen codons have a KSS above 0.80, ~50% of the upstream, noncanonical TICs implicated in cancers have a score above this value. The distribution of KSS for the annotated ATG TICs used for the main gene protein product in the 87 genes is also left-skewed, with a median of 0.746 (Figure 2).

Remarkably, the sequences surrounding the ATG codons used for the main protein translation tend not to be as close to the Kozak sequence as those surrounding the upstream noncanonical TICs. A one-sided Mann–Whitney U test showed that the noncanonical TICs had a higher KSS compared to the KSS of the canonical TICs (*p* = 0.027 or *p* = 0.020, assuming that the lowest noncanonical value is an outlier and is therefore removed).

### 2.2. KSS and the Identification of TICs Associated with Cancer

We assigned a rank for each TIC based on its KSS value relative to the KSS of all other noncanonical and ATG codons upstream from the main TIC used by the gene transcript. If the identified TIC has the highest KSS among all upstream near-cognate and ATG codons in the same sequence, it was assigned a rank of one, i.e., most likely to initiate translation. If it has the second-highest KSS, then the rank is two, etc. We repeated the same procedure for the annotated canonical TICs; however, we only compared these TICs to other ATGs in the same sequence [25,26]. When plotted on a graph, the distribution of ranks of upstream noncanonical TICs is visibly right-skewed (Figure 3). The ranks of noncanonical TICs are distinctly low, with a mode of 1 and median of 3. As in the case for noncanonical codons, the canonical TICs had ranks clustered at low values with a left-skewed distribution (Figure 3), a mode of 1, and a median of 5. Of an average of 31 potential initiation codons in the upstream region of transcripts of analyzed genes, the three translation codons with the highest KSS contained a noncanonical TIC in about 55% of cases. From an average of 75 potential ATG TICs in the full transcript of these genes, the top five codons with the highest KSS contained the canonical translation initiation codon in 54% of cases. Overall, more TICs were assigned a KSS rank of 1 than any other rank. The low value of the rank of noncanonical TICS and canonical TICs are striking when compared to an average of 31 and 75 codons present in the 5′UTR and full gene transcript, respectively. In a few cases, however, the KSS of the annotated TIC was low compared to other putative codons in the same sequence. For example, in the most extreme case, one of the annotated TICs had a KSS that was less than the KSS of 193 ATGs in the same sequence. Of note, some genes have a second identified upstream translation initiation codon in their mRNA. In the latter case, the rank value of one of the two TICs must be lower than 1 since no more than one TIC can be ranked 1 from the same sequence. The results of the analysis show that KSS effectively identifies TICs.

### 2.3. Proximity to the mRNA 5′ Terminus Is Not a Determinant of Noncanonical Translation Initiation

According to the leaky scanning model of translation initiation in eukaryotes, important factors that favor translation initiation include the proximity of the translation-initiating codon to the 5′ end of the transcript as well as an appropriate nucleotide context flanking the codon [27]. While these statements are true for ATG codons, our results show that codon position is not a determinant of TIC selection of noncanonical codons upstream of the main translation-initiating codon. In the present study, 55 of the 87 genes (63%) had a canonical TIC as the ATG that was nearest to the 5′ end of the mRNA. In contrast, however, the median number of ATGs and noncanonical codons upstream of the identified noncanonical TIC is 13. Whereas translation initiation from canonical ATGs tends to prefer the first such codon in a transcript, initiation from noncanonical TICs is less stringent. Figure 4 shows the position of upstream noncanonical TICs in addition to the position of the canonical TICs used for translation of the main protein, with the position relative to the 5′UTR and the full mRNA sequence. Noncanonical TICs identified for *EPB41L3* and *SEPTIN9*, which are downstream of the main, canonical TIC of the gene, are not included in Figure 4. Although transcript length varies from 510 to 14,805 nucleotides with a median of 3351 nucleotides, the furthest that a canonical TIC is annotated is 1287 nucleotides from the start of a transcript. Of note, noncanonical TICs are present across all regions of the 5′UTR. 

We speculate that the reason that position in the 5′UTR appears less important for the selection of an upstream noncanonical TIC may relate to the fact that a large number of ribosomes scan but remain unbound to the transcript in this upstream region. These ribosomes may complex with noncanonical TICs, particularly near-cognate codons that have a favorable nucleotide context. On the other hand, if ribosomes find a canonical TIC in good context, significantly fewer will be available to complex with ATG and noncanonical codons in the remainder of the coding region. In fact, at least 20 times as many ribosomes attach to a canonical TIC compared to a noncanonical TIC [25]. A favorable KSS of a canonical TIC may be the reason that alternative translation initiation is unlikely to occur downstream of the main TIC used for the gene protein. We observed no trend regarding the distance between the upstream noncanonical TIC and the canonical TIC used for translation of the gene’s main protein.

### 2.4. Cancer-Associated Genes with Alternative Upstream TICs Have a Longer 5′UTR Than Genes Not Associated with Cancer

We questioned whether the mRNAs investigated in this study that had upstream TICs and links to cancer tend to have a long 5′UTR. Since upstream TICs can initiate translation from any position in the 5′UTR (Figure 4), a longer 5′UTR may be more conducive to upstream translation initiation events. We compared 5′UTR length between the 85 transcripts with upstream TICs (excluding *EPB41L3* and *SEPTIN9*) with the mRNA sequences of 3615 genes that had no recorded link to cancer. We did not use other cancer-associated genes outside of this study. Remarkably, the cancer-associated genes had a statistically significant longer 5′UTR than non-cancer-associated genes according to a one-sided Mann–Whitney U test (*p*-value = 9.96 × 10^−10^). Compared to a median 5′UTR length of 205 nucleotides for genes associated with cancer, the genes not associated with cancer had a median 5′UTR length of 112 nucleotides (Figure 5). In summary, it appears that genes with a longer 5′UTR tend to be upregulated in cancer and have upstream noncanonical TICs. This finding raises the possibility that some genes with long 5′UTRs but without a recorded association with cancer may be found to have such an association in the future.

The observed tendency for cancer-associated genes to have longer 5′UTRs than typical suggests that genes with a long 5′UTR are more likely to be cancer-associated. In addition, genes are less likely to be associated with cancer as the length of the 5′UTR decreases.

### 2.5. Use of the KSS Algorithm to Identify TICs in Other Cancer Genes

One question that arises is whether the KSS algorithm can correctly identify experimentally verified alternative TICs used in cancer in genes unrelated to the 87 we analyzed. As shown below, we focused on three genes that are associated with cancer: *MAPKAPK2* [28,29,30], *ATF4* [31,32], and *BCL2* [33].

*MAPKAPK2* has two noncanonical TICs upstream of the annotated TIC [34]. We analyzed a transcript of *MAPKAPK2* (Nucleotide accession: NM_004759.5) using the KSS algorithm, and selected for near-cognate and ATG codons (Figure 6). Codons are color-coded blue, teal, green, orange, or red to reflect low to high KSS following that color order, i.e., blue has very low KSS while red has the highest KSS. The highest KSS score identifies the CTG TIC upstream of the ATG that initiates translation of the main gene product; however, a GTG is also a TIC that is used for the upstream translation, even though it has the sixth-highest KSS score.

Two TICs have been identified in the mouse ortholog of *ATF4* that are located upstream of the TIC used for translation of the main gene protein [35]. Importantly, both upstream open reading frames are conserved in human sequences [35]. Part of a transcript of this gene (Nucleotide accession: NM_009716.3) containing the TICs was analyzed for ATG codons by the KSS algorithm (Figure 7). Three experimentally identified TICs with the highest KSS in the ATF4 transcript are in red. Of note, the two upstream ATG TICs have a higher KSS than the ATG used for translation of the main gene product. 

All isoforms of *BCL2* have a number of ATGs upstream of the ATG that initiates translation of the main gene product (Figure 8). An internal ribosome entry site (IRES) mediates translation initiation of the latter ATG [36] (Nucleotide accession: XM_047437733.1). Furthermore, this initiating ATG has a higher KSS than other ATG codons upstream and within the coding region, as well as in other isoforms of the BCL2 transcript that have varying numbers of ATGs upstream from the annotated translation initiation codon. The low KSS of ATGs upstream of the ATG that initiates translation of the main gene product suggests that there may be no canonical translation initiated upstream of this main ATG, although upstream noncanonical TICs are a possibility.

### 2.6. Identifying Potential Upstream TICs in Cancer Genes

In addition to the genes analyzed above, we used the KSS algorithm to identify potential TICs in the 5′UTR of transcripts of 48 genes associated with cancer and overexpressed in solid tumors [37] (Appendix A).

## 3. Materials and Methods

### 3.1. Mapping TICs 

A recently published study that employed ribosome profiling and mass spectroscopy provided noncanonical TICs upstream of the TICs used for main protein translation [14]. The study also detailed the amino acid sequences of peptides translated in vitro from the upstream region of the genes as well as the database accession numbers for most of their associated gene transcripts. Python code was then used to: (a) retrieve each nucleotide sequence from the NCBI Nucleotide or the Ensembl database via their accession numbers, (b) translate possible reading frames of each sequence, (c) and, finally, find regions that overlap with the proteins uncovered in the study. The KSS of the TICs was then calculated using the 10 flanking nucleotides on each side of the codon. The TICs of two genes, RANBP2 and RGPD6, had ten nucleotides preceding and following the initiating GTG codon that were identical. 

Polypeptide sequences were not included if they could not be mapped to mRNA transcripts in the Nucleotide database. Nearly all unique sites mapped from the ribosome profiling data overlapped with sites mapped from the mass spectroscopy data. Of note, we excluded mass spectroscopy data whenever both of the following were true: (a) the acetylated peptide did not have methionine in the N-terminal position, (b) there was no ATG or near-cognate codon as the very next upstream codon of the nucleotide sequence to which the peptide was mapped. A total of 29 data instances were excluded because they either met these criteria or could not be mapped to an mRNA transcript annotated in the Nucleotide database. We used these criteria because of issues related to proteolytic cleavage that occurs with mass spectroscopy, as follows [14]. In some cases, the N-terminal methionine is cleaved and perhaps a few more amino acids during or following translation in eukaryotes [14,17]. Since methionine is usually the first amino acid translated [14,38], an acetylated peptide that does not have a methionine in this position likely had methionine cleaved after translation initiation. If the next codon upstream of the position to which the peptide is mapped is near-cognate or ATG, we assume that the codon is the TIC. If the upstream codon is not near-cognate or ATG, then it is likely that additional amino acids besides methionine are cleaved, and therefore the data are discarded. Translation initiation from noncanonical codons that are not near-cognate is unlikely due to ribosome destabilization at the codon, since codons that diverge by even one nucleotide from ATG have much less stable ribosome base pairing, making translation initiation much less energetically favorable [26]. All peptides we analyzed in this study that had methionine in the N-terminal position mapped to ATG or near-cognate TICs. 

### 3.2. Randomized Sampling of Codons to Establish a KSS Baseline

A selection of random codons in this study served as a baseline for the KSS distribution (Figure 2). The random codons were obtained from Entrez eSearch [39] by retrieving 200,000 accession numbers of the total ~9.1 million accessions for annotated human mRNA in the NCBI Nucleotide database. Of note, Entrez cannot return accessions at random, and therefore consistently fetches accessions in the same order by an arbitrary measure of relevance. For this reason, we randomly sampled 25,000 accession numbers of the 200,000 total. We then retrieved the mRNA sequences of the 25,000 accessions via Entrez eFetch, and randomly selected four codons with flanking sequences from each transcript to compute the KSSs.

### 3.3. Retrieving Sequences Not Associated with Cancer

The Ensembl FTP tool pulled data for all genes currently annotated in the human genome (from the GRCh38 assembly). The names of 19,349 genes listed as protein-coding were extracted. Next, Entrez eSearch provided the number of publications for each gene in PubMed that contained the gene name as well as the word “cancer” in either the title or abstract; 3725 genes had no recorded link to cancer mentioned in the titles or abstracts of published literature. Entrez eSearch then retrieved accession numbers for mRNA sequences of these remaining genes; accessions of predicted mRNA sequences were excluded. eFetch was then used to retrieve the actual sequences of the mRNA from NCBI Nucleotide, and one mRNA sequence was kept per gene for analysis. In total, 3615 of the 3725 genes without a recorded link to cancer had a confirmed mRNA sequence in the database and were analyzed.

## 4. Discussion

The results of this study indicate that the KSS algorithm is effective at identifying both noncanonical and canonical TICs in cancer genes upstream of the ATG used for translation of the main gene protein. The KSS algorithm frequently narrowed the possible location of the TIC of the alternative and main gene product to one codon. In other cases, the KSS significantly reduced the number of codons that could initiate translation per gene. It is important to note that the analyzed genes may still have additional noncanonical or ATG TICs that have not yet been identified.

The results of the present study show that nearly all upstream TICs associated with cancer have flanking nucleotides that closely match the Kozak sequence. In fact, the upstream noncanonical TICs have a statistically significant better match to the Kozak sequence than the canonical codons used for translation of the main protein of the gene. Contrary to the leaky scanning model of translation initiation for ATGs [27], the proximity of the noncanonical TIC to the 5′ end of the transcript does not appear to be a significant factor in determining whether the codon is used for translation initiation. It appears that the similarity of sequences flanking the noncanonical codon to the Kozak sequence has more predictive value than its position in the sequence.

Because of the success of the KSS algorithm in identifying upstream TICs used in cancer, we employed this same algorithm to make predictions about upstream TICs in other cancer genes (Appendix A). The KSS algorithm can be useful in this regard.

Importantly, we found that the mRNA from cancer-associated genes had substantially longer 5′UTRs than genes not associated with cancer. This finding raises the question of whether longer 5′UTRs are a defining characteristic of many genes overexpressed in cancer. The longer 5′UTR presents more opportunities for ribosome binding at upstream noncanonical TICs, which may drive tumor formation [11].

It is possible that some near-cognate codons will very rarely initiate translation irrespective of their KSS (Figure 1), likely due to instability of ribosome base pairing to such codons [26,40]. Ribosome base pairing with CTG, GTG, ACG, and TTG is most stable, causing these codons to be more likely to initiate translation than other near-cognate codons [40]. This may explain why upstream translation initiation occurs from both the highest (CTG) and the sixth-highest (GTG) KSS-scoring codons in *MAPKAPK2* (Figure 6). If only most common near-cognate codons (GTG, CTG, ACG, and TTG) are considered in the *MAPKAPK2* 5′UTR, the GTG TIC has the second-highest KSS following the highest-scoring CTG TIC. Thus, isolating the most commonly used initiation codons for analysis (i.e., CTG, GTG, ACG, TTG) with the KSS algorithm may produce even higher accuracy in narrowing TICs. We provide this functionality, as our KSS tool available online allows users to select particular codons for analysis.

A limitation in this study is that other factors were not assessed that may enhance noncanonical translation initiation in oncogenic genes. For example, secondary structure of the mRNA might impact translation initiation from codons that do not have a high KSS [9]. Still, a high KSS along with an optimal secondary structure may enhance translation initiation more than the secondary structure alone. This may, for example, be the case for the *BCL2* TIC, which has an IRES as part of the secondary structure as well as a high KSS.

In summary, the KSS algorithm appears to be an effective tool for the identification of TICs associated with cancer. Importantly, this machine learning algorithm can predict noncanonical and canonical TICs in mRNA sequences [12].

An interactive KSS calculator that computes scores in input nucleotide sequences is available at https://www.tispredictor.com/kss (accessed on 20 August 2022).

## Figures and Tables

**Figure 1 ijms-23-10564-f001:**
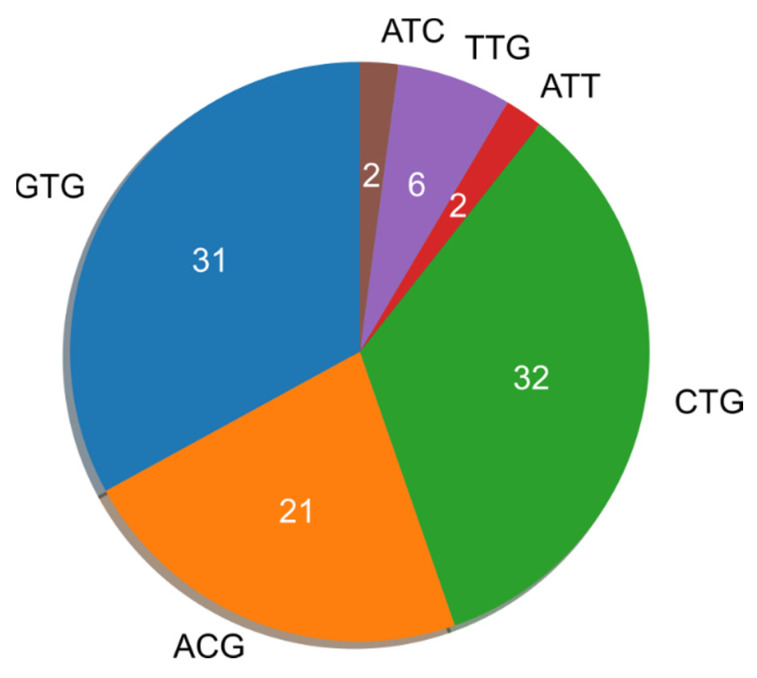
Noncanonical TICs from selected cancer genes.

**Figure 2 ijms-23-10564-f002:**
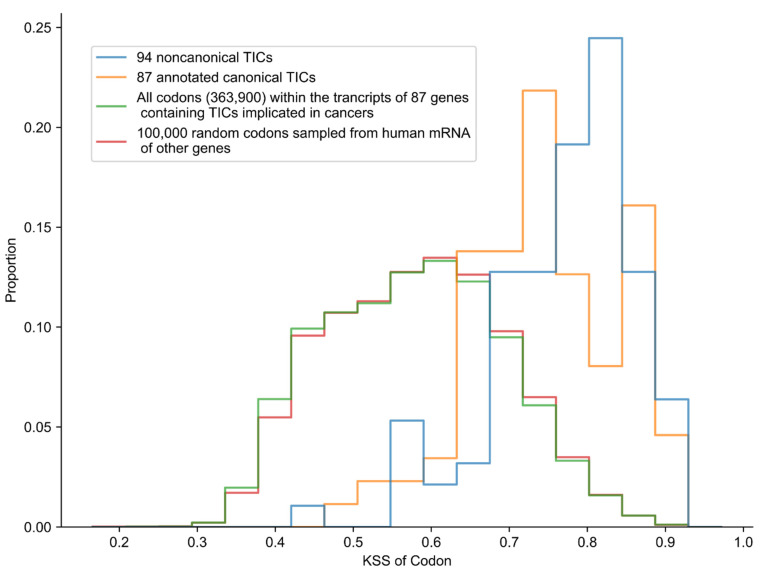
KSS distribution of codons.

**Figure 3 ijms-23-10564-f003:**
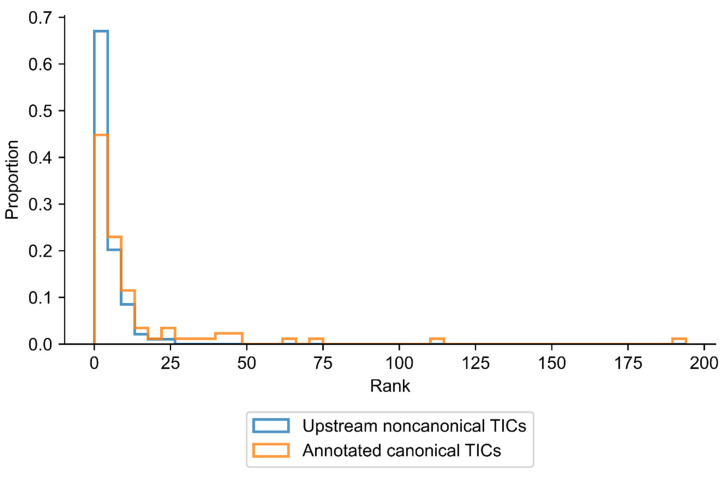
Rank of TICs in genes associated with cancer.

**Figure 4 ijms-23-10564-f004:**
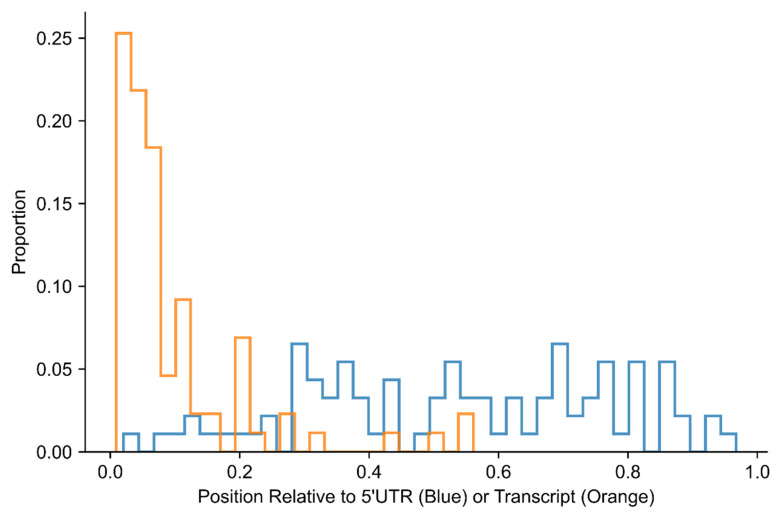
Position of upstream noncanonical TICs in the 5′UTR (blue) and canonical TICs in the mRNA transcript (orange).

**Figure 5 ijms-23-10564-f005:**
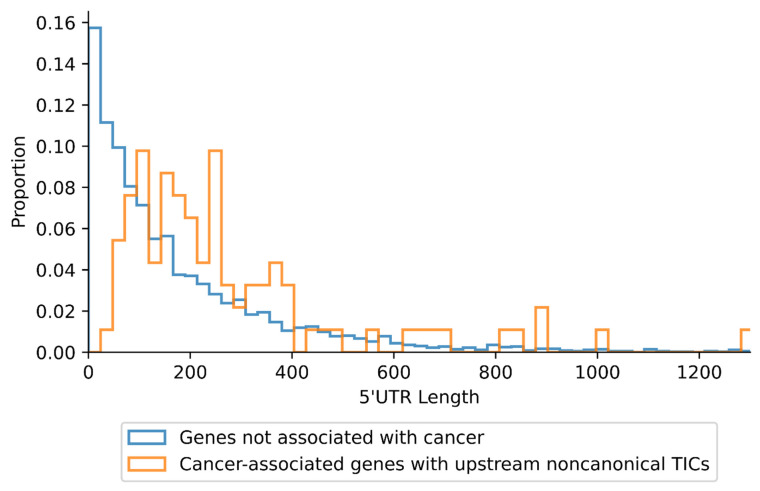
The length of the 5′UTR in genes that are not associated with cancer as well as in cancer-associated genes with upstream noncanonical TICs.

**Figure 6 ijms-23-10564-f006:**
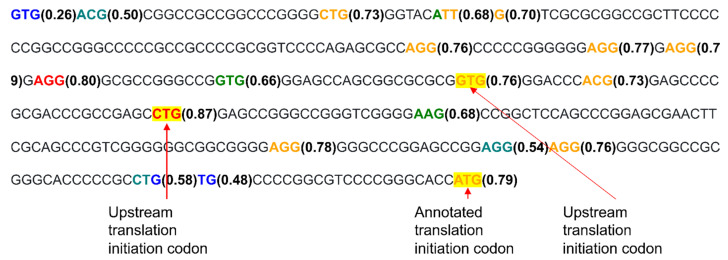
5′UTR of *MAPKAPK2* transcript with near-cognate and ATG codons color-coded. In this figure and subsequent ones, the KSS is in parentheses after the codon.

**Figure 7 ijms-23-10564-f007:**
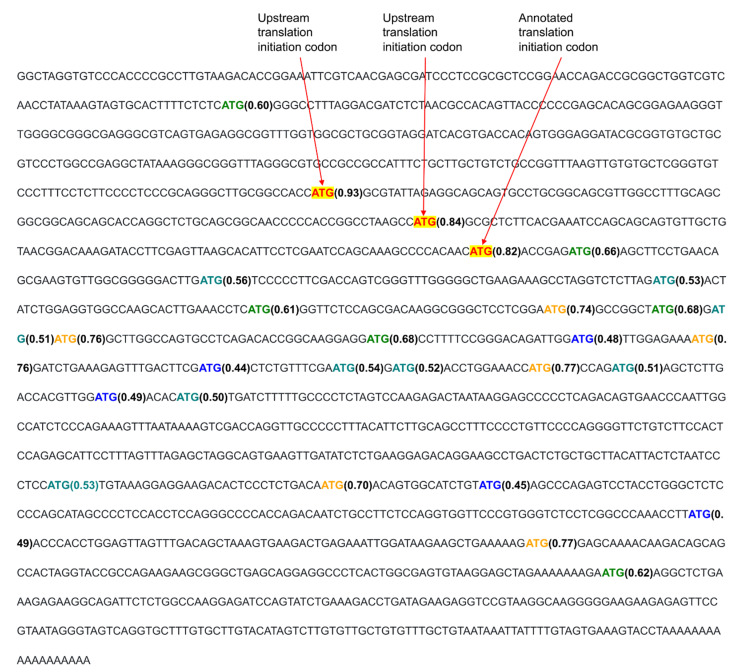
Mouse *ATF4* transcript with ATG codons color-coded.

**Figure 8 ijms-23-10564-f008:**
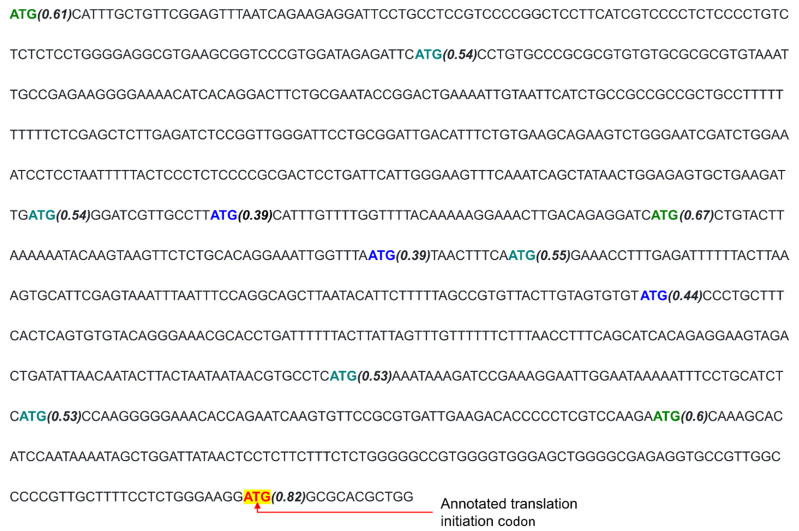
Part of a transcript of *BCL2* isoform X1 with color-coded TICs and ATG codons. The TIC shown in red has a higher KSS than all ATGs in comparable parts of *BCL2* isoform transcripts.

## Data Availability

The code to reproduce findings is accessible at https://github.com/Agleason1/Alternative-TICs-Implicated-in-Cancers/ (accessed on 20 August 2022). A DOI was assigned to the repository using Zenodo: http://doi.org/10.5281/zenodo.6987364 (accessed on 20 August 2022).

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
