# Peer review of "Kozak Similarity Score Algorithm Identifies Alternative Translation Initiation Codons Implicated in Cancers"

_ijms, 2022, doi:10.3390/ijms231810564_

Round 1
Reviewer 1 Report
KSS algirithm can successfully predict TIC in cancer associated genes.
However, as pointed out by authors, the study has many limitations:
1. Did authors have any evidence about the rate of effective translation rate starting from alternative TICs in respect to the canonical one?
2. Authors have to divide in 2 categories their genes: the ones with IRES and the ones without and compare the likelood of successfull prediction.
3. Please provide the scatter dot plot for the lenght of 5'utr of cancer associated genes and not cancer associated ones.
4. A comparison with tumor suppressors mutated in cancer can be useful, eg TP53, APC, PTEN and the ones of DDR. Any evidence about a burst of translation starting from identified TICs?
5. Which is the rate of mutations of those alternative TICs in cancer?
Reviewer 2 Report
The manuscript “Kozak Similarity Score algorithm identifies alternative translation initiation codons implicated in cancers” by Gleason and co-authors demonstrate an applicability of Kozak similarity score search to find non-canonical translation initiation sites in cancer-related genes.
Authors analyzed canonical and non-canonical TIS to find that non-canonical initiation sites are on average closer to the Kozak consensus. This is no surprise, since for initiation to happen, a deviation of the initiation codon from AUG should be compensated, otherwise such site would not be used for initiation. Additionally, authors demonstrated that non-canonical initiation sites are distributed more widely and do not have a tendency to be located closer to the 5’-end. Finally, authors demonstrated that cancer-related mRNAs have on average longer 5’-UTRs and it is possible to find sites with different Kozak similarity scores in those 5’-UTRs. So what? It is likely, that cancer related mRNAs have been evolved to minimize the probability of cancer and longer 5’-UTRs with non-canonical translation intuition sites aim to decrease translation of the main open reading frame. It would be interesting to compare some numerical value of the propensity to find non-canonical initiation sites in 5’-UTRs of cancer-related genes and other genes, normalized to the same length. It would be useful to check whether predictions made by authors could be supported by the data. The example provided at the Figure 5 contradicts the data (https://gwips.ucc.ie/cgi-bin/hgTracks?db=hg38&lastVirtModeType=default&lastVirtModeExtraState=&virtModeType=default&virtMode=0&nonVirtPosition=&position=chr1%3A206684967%2D206685148&hgsid=287567_j7En4o0aA50HTeqa84SOH6FMYp36). You may see, that predicted codon is not used for initiation, but rather a different one is really utilized. The predictions should be systematically verified. Additionally, nothing is discussed about the products of translation started at those non-canonical start sites. Are they functional? Regulatory of the main ORF? Cancer-related?
Round 2
Reviewer 1 Report
Please introduce comments about IRES and TIC in the discussion and the difficulties in predicting IRES usage
Reviewer 2 Report
My concernes have been addressed.